# Orbital Edelstein effect from the gradient of a scalar potential

I. A. Ado[1⋆], M. Titov[1], Rembert A. Duine[2,3] and Arne Brataas[4]

**1** Radboud University, Institute for Molecules and Materials,
Heyendaalseweg 135, 6525 AJ Nijmegen, The Netherlands
**2** Institute for Theoretical Physics, Utrecht University,
Princetonplein 5, 3584 CC Utrecht, The Netherlands
**3** Department of Applied Physics, Eindhoven University of Technology,
P.O. Box 513, 5600 MB Eindhoven, The Netherlands
**4** Center for Quantum Spintronics, Department of Physics, Norwegian University of Science and
Technology, Høgskoleringen 5, 7491 Trondheim, Norway
⋆ iv.a.ado@gmail.com

October 2, 2024

## Abstract

**We study the orbital Edelstein effect (OEE) that originates from a particular inversion symmetry breaking mechanism: an asymmetric scalar potential. We compute OEE of this kind with the help of the Kubo formula in the diffusive regime for a parabolic band Hamiltonian. We also present a qualitative derivation of the effect. Both approaches give the same result. This result does not rely on spin-orbit coupling (SOC) and scales as a cube of the momentum relaxation time. In sufficiently clean large systems with weak SOC, OEE of this nature should exceed the spin Edelstein effect by orders of magnitude. It may also provide an alternative interpretation for some experiments concerning the spin Hall effect.**

# 1 Introduction

In nonmagnetic time-reversal invariant systems, an external electric field can generate nonequilibrium magnetization. If only the spin magnetic moments contribute to the latter, the corresponding effect is usually called the spin Edelstein effect (SEE) [1]. In the past decades, it attracted significant attention as a way to manipulate magnetic properties of materials by electrical means, in particular in the form of spin-orbit torques [2, 3]. The later terminology reflects the fact that SEE is a relativistic effect and requires spin-orbit coupling (SOC).

In recent years, the focus of research partly shifted from spin to orbital contributions to different magnetic phenomena, both equilibrium and nonequilibrium [4–6]. Importantly, nonequilibrium magnetic effects of the orbital nature are often believed to be stronger than their spin counterparts as the former do not necessarily rely on SOC. The orbital Edelstein effect (OEE) represents a particular example of such an effect [7, 8]. It originates from the reaction of the orbital magnetic moments of the particles (rather than the spin ones) to the external electric field.

For finite OEE, inversion symmetry of the system should be broken [5, 9]. In the past, OEE was studied for the following symmetry breaking mechanisms: chiral hoppings [7], *sp* hybridization at metal surfaces [10], bilayer system asymmetry [11], interaction-induced charge density wave order [12], quadrupolar symmetry breaking [13], and, possibly, others. The more general magnetoelectric effect was also computed for asymmetric scattering from impurities [14, 15].

However, one particular source of asymmetry that is very common in spintronics has been overlooked in the studies of OEE so far, namely the asymmetric scalar potential. To be more precise, possible effects of such a potential on OEE were considered only via the SOC term generated by it, while the potential itself was disregarded. At the same time, its gradient translates into an internal electric field that can be large both in bulk materials [16, 17] and at interfaces [18, 19]. This internal field, as we demonstrate in the present work, gives rise to a strong OEE and cannot be ignored in this regard.

Below, we consider an infinite three-dimensional conducting system which is translationally invariant in the $xy$ plane and is modeled by a parabolic band Hamiltonian

$$H = \frac{p^2}{2m} + U(z), \qquad p^2 = p_x^2 + p_y^2 + p_z^2, \qquad p_\alpha = -i\hbar\nabla_\alpha, \tag{1}$$

with an isotropic effective mass $m$. For simplicity, the internal asymmetric scalar potential $U(z)$ is approximated by a linear function:

$$U(z) = \frac{\partial U}{\partial z}z, \qquad \frac{\partial U}{\partial z} = const. \tag{2}$$

A time-dependent external electric field $E_x$ is applied to this system along the $x$ direction and OEE is recognized as a linear in $E_x$ contribution to the nonequilibrium average density $M_y$ of the orbital magnetic moment operator in the $y$ direction. We compute $M_y$ in the diffusive regime, thus the linear approximation for $U(z)$ is allowed.

The chosen model can be interpreted, e.g., as a simplified description of a bulk Rashba material [16, 17, 20–22], in which we completely ignore SOC. For our purposes, the most important characteristic of the system is the finite gradient of the scalar potential in Eqs. (1), (2), which any Rashba material provides. We also assume that $\partial U/\partial z \neq 0$ translates into a finite gradient of the particle density. The latter is just the condition of equilibrium with respect to the $z$ direction.

The mechanism of OEE in the considered model, in fact, can be understood without any complex computations. Assume that the external field $E_x$ forces electrons to move with the drift velocity $v_x = v_{\text{drift}}$. The total orbital magnetization $M_y$ is then proportional to the average value of $zv_x$. If this value is understood as a spatial average weighed with the particle density $n(z)$, the gradient of the latter in the $z$ direction ensures that an uncompensated net orbital magnetization is created. The detailed version of this qualitative analysis is presented in Sec. 2. A full microscopic computation of OEE is performed in Sec. 3. In Sec. 4, we discuss the results.

## 2 Elementary derivation

### 2.1 Density distribution

First, let us establish the relation between the density distribution and the scalar potential gradient. The equilibrium condition for the system is

$$\mu(z) + U(z) = \mu(0) = \mu_0 = const, \tag{3}$$

where $\mu_0$ is the electrochemical potential. Note that the chemical potential $\mu(z)$ varies in the $z$ direction and that the Fermi-Dirac distribution function depends on $\mu_0$ rather than on $\mu(z)$:

$$f_\varepsilon = \Theta(\mu_0 - \varepsilon), \tag{4}$$

where we have assumed zero temperature. We introduce the retarded and advanced Green's functions for the Hamiltonian of Eq. (1), $G^{R,A} = (\varepsilon \pm i0 - H)^{-1}$, and expand them with respect to the scalar potential gradient:

$$G^{R,A} = G_0^{R,A} + \frac{\partial U}{\partial z} G_0^{R,A} z\, G_0^{R,A}, \qquad G_0^{R,A} = (\varepsilon \pm i0 - H_0)^{-1}, \qquad H_0 = \frac{p^2}{2m}. \tag{5}$$

Here $H_0$ describes the homogeneous system and we disregarded higher-order terms in the Dyson series.

The particle density at the position $\boldsymbol{r}$ reads

$$n(\boldsymbol{r}) = \frac{\gamma_s}{2\pi i} \int d\varepsilon f_\varepsilon \left[ G^A(\boldsymbol{r},\boldsymbol{r}) - G^R(\boldsymbol{r},\boldsymbol{r}) \right], \tag{6}$$

where $\gamma_s = 2$ is the spin degeneracy factor. Next, we use the expansion of Eq. (5) and the fact that $G_0^{R,A}$ correspond to a fully translationally invariant system. This allows us to switch to the momentum representation in Eq. (6),

$$n(\boldsymbol{r}) = \frac{\gamma_s}{2\pi i} \int \frac{d^3 p}{(2\pi\hbar)^3} d\varepsilon f_\varepsilon \left( \left\{ G_0^A(\boldsymbol{p}) - G_0^R(\boldsymbol{p}) \right\} + z\frac{\partial U}{\partial z} \left\{ \left[ G_0^A(\boldsymbol{p}) \right]^2 - \left[ G_0^R(\boldsymbol{p}) \right]^2 \right\} \right), \tag{7}$$

with $G_0^{R,A}(\boldsymbol{p}) = 1/\left( \varepsilon \pm i0 - p^2/2m \right)$. Computing the momentum integrals by the residue theorem and employing Eq. (4) to integrate over energy, we find for the particle density:

$$n(z) = n_0 \left( 1 - \frac{3}{2} \frac{z}{\mu_0} \frac{\partial U}{\partial z} \right), \qquad n_0 = n(0) = \frac{\gamma_s \sqrt{2(m\mu_0)^3}}{3\pi^2\hbar^3}. \tag{8}$$

The expression for $n_0$ is of course well-known.

## 2.2 Magnetization as a spatial average

To compute the orbital magnetization, we need to define the orbital magnetic moment operator. The Hamiltonian of Eq. (1) in the presence of a magnetic field $\boldsymbol{B} = B_y \boldsymbol{e}_y$ can be expressed as

$$H_B = \frac{1}{2m} \left( \boldsymbol{p} - \frac{e}{c} \boldsymbol{e}_x B_y z \right)^2 + U(z),$$  (9)

where $e$ is the electron charge, $c$ is the speed of light, $\boldsymbol{p} = -i\hbar\boldsymbol{\nabla}$, and we disregarded the Zeeman term since it is not important for OEE. The $y$ component of the orbital magnetic moment operator then reads

$$\mathcal{M}_y = -\frac{\partial H_B}{\partial B_y} = -\frac{2\mu_B m_0}{\hbar} z v_x, \qquad \mu_B = -\frac{e\hbar}{2m_0 c},$$  (10)

where $\mu_B$ is the Bohr magneton, $v_x = p_x/m$, and $m_0$ is the electron mass in vacuum.

We would like to compute the average density of the observable $\mathcal{M}_y$. The result does not depend on the position, and thus we compute it at $\boldsymbol{r} = 0$. We assume that only the regions at distances of the order of the mean free path $l$ contribute to the result because at larger distances the coherence is lost due to scattering from impurities. For $v_x$, we substitute the drift velocity $v_{\text{drift}} = eE_x\hbar\tau/m$, where $\tau$ is the transport relaxation time measured in the inverse energy units. In this work, we consider isotropic scattering from delta-correlated scalar impurities, hence $\tau$ coincides with the scattering time. Under the assumptions made and using Eq. (10), we write

$$M_y = \frac{[\mathcal{M}_y]_{V_l}}{V_l} = -\frac{2\mu_B m_0}{\hbar} \frac{eE_x\hbar\tau}{m} \frac{1}{V_l} \int\limits_{V_l} d^3r\, n(z)z,$$  (11)

where $[\mathcal{M}_y]_{V_l}$ denotes the total orbital magnetic moment "contained" in the cube $V_l$. The latter is centered at $\boldsymbol{r} = 0$, and the length of its edges is equal to $2l$. In Eq. (11), $d^3r\, n(z)$ expresses the total number of particles in the volume $d^3r$. Therefore, $M_y$ in this formula has a meaning of the orbital magnetization averaged over all particles in $V_l$.

Once Eq. (8) is substituted in Eq. (11), we integrate over $z$ and find

$$M_y = \frac{\mu_B m_0}{\hbar} \frac{eE_x\hbar\tau}{m} \frac{n_0}{\mu_0} \frac{\partial U}{\partial z} l^2.$$  (12)

This result can also be expressed in terms of the charge current density $j_x = en_0(eE_x\hbar\tau/m)$:

$$M_y = \frac{\mu_B m_0}{e\hbar} j_x \frac{l^2}{\mu_0} \frac{\partial U}{\partial z} = -\frac{1}{2c} j_x \frac{l^2}{\mu_0} \frac{\partial U}{\partial z}.$$  (13)

We note that, since $l \propto \tau$ and $j_x \propto \tau$, the computed $M_y$ is of the third order with respect to $\tau$.

## 3 Microscopic derivation

### 3.1 Self-energy

Let us now derive the results of Eqs. (12), (13) microscopically. For this, we supplement the Hamiltonian of Eq. (1) with a scalar Gaussian delta-correlated disorder potential with zero mean,

$$H_{\text{dis}} = \frac{p^2}{2m} + U(z) + U_{\text{dis}}(\boldsymbol{r}), \quad \langle U_{\text{dis}}(\boldsymbol{r}_1)U_{\text{dis}}(\boldsymbol{r}_2)\rangle = \alpha_{\text{dis}}\delta(\boldsymbol{r}_1 - \boldsymbol{r}_2), \quad \langle U_{\text{dis}}(\boldsymbol{r})\rangle = 0,$$  (14)

where the angular brackets denote the ensemble average. In the dc limit, at zero temperature, the orbital magnetization $M_y$ can be expressed with the help of the Kubo-type [23] formula,

$$M_y = \gamma_s \frac{eE_x\hbar}{2\pi V} \left\langle \text{Tr}\left[ v_x G_{\text{dis}}^A \mathcal{M}_y G_{\text{dis}}^R \right] \right\rangle, \qquad G_{\text{dis}}^{R,A} = (\mu_0 \pm i0 - H_{\text{dis}})^{-1},$$  (15)

where $V$ is the system volume, $v_x = p_x/m$, and, due to $\partial f_\varepsilon / \partial \varepsilon = -\delta(\varepsilon - \mu_0)$, the energy argument of the Green's functions' is fixed at $\mu_0$ (from now on).

We compute $M_y$ in the noncrossing approximation [24, 25] as we are interested only in the contributions of the leading order with respect to $\tau$[1]. In this case, averaging in Eq. (15) boils down to adding diffusion ladder vertex corrections to either $v_x$ or $\mathcal{M}_y$ and to replacing each Green's function with its average. For our simple model, all vertex corrections vanish[2], thus we only need to average the Green's functions. In the first Born approximation, we have

$$\langle G_{\text{dis}}^{R,A} \rangle = \left( \mu_0 - \Sigma^{R,A} - H \right)^{-1}, \qquad \Sigma^{R,A}(\mathbf{r}) = \alpha_{\text{dis}} G^{R,A}(\mathbf{r}, \mathbf{r}), \tag{16}$$

or, neglecting the real part of the self-energy $\Sigma^{R,A}$,

$$\Sigma^R(\mathbf{r}) = -\Sigma^A(\mathbf{r}) = \frac{1}{2} \left[ \Sigma^R(\mathbf{r}) - \Sigma^A(\mathbf{r}) \right] = \frac{\alpha_{\text{dis}}}{2} \left[ G^R(\mathbf{r}, \mathbf{r}) - G^A(\mathbf{r}, \mathbf{r}) \right]. \tag{17}$$

Comparing this with Eq. (6) and using Eq. (8), we find

$$\Sigma^R(\mathbf{r}) = -\frac{i\pi \alpha_{\text{dis}}}{\gamma_s} \frac{\partial n(\mathbf{r})}{\partial \mu_0} = -\frac{i}{2\tau} \left( 1 - \frac{z}{2\mu_0} \frac{\partial U}{\partial z} \right), \qquad \tau = \frac{\pi \hbar^3}{\alpha_{\text{dis}} \sqrt{2m^3 \mu_0}}. \tag{18}$$

In the following, we disregard the position-dependent part of the self-energy because its contribution to $M_y$ is smaller by the factor $1/(\mu_0 \tau)$ than the leading order one. Hence the ensemble average of the Green's functions is provided by the expression

$$g^{R,A} = \langle G_{\text{dis}}^{R,A} \rangle = \left( \mu_0 \pm \frac{i}{2\tau} - H \right)^{-1}, \tag{19}$$

for which we introduced a short-handed notation $g^{R,A}$.

## 3.2 Magnetization as a statistical average

We can now see that, after averaging, the Kubo formula of Eq. (15) reads

$$M_y = \gamma_s \frac{eE_x \hbar}{2\pi V} \text{Tr} \left[ v_x g^A \mathcal{M}_y g^R \right]. \tag{20}$$

To evaluate it, one should use an expansion analogous to that of Eq. (5),

$$g^{R,A} = g_0^{R,A} + \frac{\partial U}{\partial z} g_0^{R,A} z g_0^{R,A}, \qquad g_0^{R,A} = \left( \mu_0 \pm \frac{i}{2\tau} - H_0 \right)^{-1}, \qquad H_0 = \frac{p^2}{2m}. \tag{21}$$

We substitute Eq. (21) into Eq. (20) and employ the definition of Eq. (10). The part independent of $\partial U / \partial z$ vanishes in the obtained expression as it corresponds to an inversion symmetric system. We are therefore left with

$$M_y = -\gamma_s \frac{2\mu_B m_0}{\hbar} \frac{eE_x \hbar}{2\pi V} \frac{\partial U}{\partial z} \text{Tr} \left[ v_x g_0^A z g_0^A z v_x g_0^R \right] + \text{c.c.} \tag{22}$$

We would like to evaluate Eq. (22) in the momentum representation. Both position operators here can be, in principle, "regularized" by means of the substitution $z \to (\sin zs)/s$ with $s$ to be sent to zero in the end. The same substitution is, in fact, used to derive the "modern theory of orbital magnetization" [26]. Although the usage is indirect in that case because the sine is introduced from the start to define the vector potential. As a result, position operators never appear in the derivations that follow. Technically, however, it does not matter, at which point the position operator is

---

[1]For the same reason we neglected all contributions in Eq. (15) that upon averaging become subleading for $\tau \to \infty$.
[2]Technically, they vanish as a result of the integration over the angles of momentum.

"regularized", because this does not change the principle result: in momentum space, the position operator $\boldsymbol{r}$ differentiates all operators to the right of it[3] with respect to $\boldsymbol{p}$[4]. For the "modern theory of orbital magnetization", this can be seen in its Green's functions' formulation [27].

Using the momentum space representation $i\hbar(\partial/\partial\boldsymbol{p})$ of the position operator $\boldsymbol{r}$ for an infinite system, we express the trace in Eq. (22) as an integral over momentum, integrate by parts twice, and arrive at

$$M_y = \gamma_s \frac{2\mu_B m_0}{\hbar}\frac{eE_x\hbar}{2\pi}\frac{\partial U}{\partial z}\hbar^2 \int \frac{d^3p}{(2\pi\hbar)^3}v_x^2(\boldsymbol{p})g_0^R(\boldsymbol{p})\frac{\partial}{\partial p_z}\left[g_0^A(\boldsymbol{p})\frac{\partial g_0^A(\boldsymbol{p})}{\partial p_z}\right] + \text{c.c.}, \quad (23)$$

where $g_0^{R,A}(\boldsymbol{p}) = 1/\left(\mu_0 \pm i/(2\tau) - p^2/2m\right)$. We also took into account that $v_x$ does not depend on $p_z$ and that all operators inside the integral commute with each other. With the help of partial integration and the relations $g_0^R g_0^A = i\tau(g_0^R - g_0^A)$ and $\partial g_0^{R,A}/\partial p_z = g_0^{R,A}v_z g_0^{R,A}$, one can bring Eq. (23) to the form

$$M_y = \gamma_s \frac{2\mu_B m_0}{\hbar}\frac{eE_x\hbar}{2\pi}\frac{\partial U}{\partial z}\frac{\hbar^2\tau^2}{m} \int \frac{d^3p}{(2\pi\hbar)^3}v_x^2(\boldsymbol{p})g_0^R(\boldsymbol{p})g_0^A(\boldsymbol{p}), \quad (24)$$

where several contributions of subleading orders with respect to $\tau$ have been omitted.

The integral here equals $2\pi n_0\tau/(m\gamma_s)$ which is, up to a prefactor, the longitudinal Drude conductivity. Therefore,

$$M_y = \frac{\mu_B m_0}{\hbar}\frac{eE_x\hbar\tau}{m}\frac{\partial U}{\partial z}\frac{\hbar^2\tau^2}{m/2}n_0. \quad (25)$$

Multiplying both the numerator and the denominator of the last fraction by the square of the Fermi velocity $v_F^2$, we reproduce the result of Eq. (12) since $\mu_0 = mv_F^2/2$ and $l = v_F\hbar\tau$.

## 4  Discussion

### 4.1  Other methods

It may be possible to compute OEE in the considered model using the theory of spatial dispersion [14, 28–30]. At the same time, it does not seem plausible that this can be done with the help of the "Boltzmann approximation" [7, 31]. Indeed, the latter produces results that are linear with respect to $\tau$, whereas our computation gives the cubic dependence.

### 4.2  Choice of the gauge

In our computation, we chose the $\mathcal{M}_y \propto zv_x$ gauge for the orbital magnetic moment operator and dismissed the $\mathcal{M}_y \propto xv_z$ gauge. This choice is determined by the spatial symmetry of the system, which is translationally invariant in the $xy$ plane and inhomogeneous in the $z$ direction due to the scalar potential $U = U(z)$. The $\mathcal{M}_y \propto xv_z$ gauge does not match this symmetry because of its explicit dependence on the $x$ coordinate. In our approach, averaging of the operator $xv_z$ over the infinite $xy$ plane is mathematically ambiguous and not well justified.

---

[3]Or to the left if one integrates by parts.

[4]For an infinite system, position operators under the trace must stand between two Green's functions. Moreover, in the momentum representation, one cannot use the cyclic permutation property to place a position operator in front of all other operators under the trace. Indeed, that would differentiate the entire integrand over the variable of integration and effectively nullify the result.

## 4.3 Linear density of the orbital magnetic moment

We derived the Kubo formula of Eq. (23) by considering the response of the operator $zv_x$ to the electric field $E_x$. This operator determines the full orbital magnetic moment of the entire system. Division of its response to $E_x$ by the system volume gives the (local) nonequilibrium orbital magnetization $M_y$ if the orbital moment is distributed uniformly in space. While the latter condition is obviously satisfied for the $xy$ plane, the distribution in the $z$ direction may, in principle, be nonuniform.

To exclude this scenario, we can, instead of considering the full orbital magnetic moment, consider its linear density. Namely, we can introduce the vector potential $A_{z_0} = e_x B_0 [\text{sign}(z - z_0)]/2$ that defines the magnetic field $B_{z_0} = e_y B_0 \delta(z - z_0)$. Then, the derivative of the Hamiltonian in the presence of such a potential with respect to $B_0$ (taken with a minus sign),

$$\delta_{z_0} \mathcal{M}_y = -\frac{\partial}{\partial B_0} \left[ \frac{1}{2m} \left( p - \frac{e}{2c} e_x B_0 \, \text{sign}(z - z_0) \right)^2 \right] = -\frac{\mu_B m_0}{\hbar} v_x \, \text{sign}(z - z_0), \quad (26)$$

defines the operator of density of the $y$ component of the orbital magnetic moment over the $z$ direction at the $z = z_0$ plane[5].

It is straightforward to demonstrate that the response of this operator to $E_x$ divided by the area of the $z = z_0$ plane is expressed by the same Eq. (23) that we derived for the response of the total orbital magnetic moment operator $\mathcal{M}_y$. As we have shown in this paper, evaluation of Eq. (23) leads to a $z$-independent result (for the considered system). Therefore, we can conclude that the spatial distribution of the orbital magnetic moment is indeed uniform and averaging of $\mathcal{M}_y$ over the volume can by replaced by averaging of $\delta_{z_0} \mathcal{M}_y$ over the plane.

## 4.4 Interfacial orbital Edelstein effect from the density gradient

In Sec. 2, we presented a qualitative interpretation of OEE originating from the density gradient. In this picture, the nonequilibrium orbital magnetization at a certain point in the material is generated by electrons at distances of the order of the mean free path $l$ away from this point. At larger distances, coherence is lost due to scattering. This "diffusive" OEE can be realized in samples with large enough sizes that exceed $l$.

We argue, however, that the same physics can occur in smaller samples, in particular at the interfaces and in heterostructures with the structural inversion asymmetry. In this case, $l^2$ in our formula for OEE would be replaced by a different squared length scale. Of course, in such systems, scattering from the boundaries can also contribute to OEE.

We note that, in large clean systems, OEE originating from the density gradient may, in principle, be characterised by colossal length scales.

## 4.5 Comparison with other effects

Generation of nonequilibrium magnetization by electric fields and charge currents in nonmagnetic materials has been extensively studied with relation to the spin Edelstein effect and the spin Hall effect (SHE). In experiment, SEE and SHE are usually probed using the Faraday and Kerr effects or by measuring torques on magnetization [2, 32]. While it is not yet clear whether OEE can produce considerable torques on magnetization, its signature in the Faraday and Kerr effects should be observable. Since OEE is not a relativistic effect, while both SEE and SHE are, we anticipate that magnetization induced due to OEE in Rashba materials can be stronger as compared with that related to SEE and SHE. In particular, this should be the case in sufficiently clean systems, in which the $\tau^3$ scaling of OEE would dominate the dependence on $\tau$ of SEE and SHE.

---

[5]Here, the notation $\delta_{z_0}$ is used to indicate that the operator $\delta_{z_0} \mathcal{M}_y$ has a meaning of density per length in the $z$ direction computed at $z = z_0$.

# 5 Conclusion

We studied the orbital Edelstein effect originating from the gradient of the scalar potential that translates into the particle density gradient. We presented a microscopic derivation of the effect and a qualitative one. Both provide the same result which does not rely on spin-orbit coupling and scales as a cube of the momentum relaxation time. We argue that the orbital Edelstein effect of this nature can be very strong in Rashba and other materials and should not be ignored in experiments on the spin Edelstein effect and the spin Hall effect.

# Acknowledgements

We are grateful to Sergii Grytsiuk and Yuriy Mokrousov for informative discussions.

**Funding information** R. A. D. has received funding from the European Research Council (ERC) under the European Union's Horizon 2020 research and innovation programme (Grant No. 725509). The Research Council of Norway (RCN) supported A. B. through its Centres of Excellence funding scheme, project number 262633, "QuSpin", and RCN project number 323766. M. T. has received funding from the European Union's Horizon 2020 research and innovation program under the Marie Skłodowska-Curie grant agreement No 873028.

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
