# Peer review of "Orbital Edelstein effect from the gradient of a scalar potential"

_SciPost Physics_

## Round 2 · Referee Report · Anonymous (Referee 1) · 2024-11-20

Strengths

1- The submitted paper addresses a topic of much current interest, the orbital Edelstein effect. 2- A new finding is that the orbital effect scales as the cube of the momentum relaxation time.

Weaknesses

1- although a part of the theory is on a high level, there can be technical questions around certain model assumptions. 2- the derived expression does not explicitly depend on the materials' specific electronic structure.

Report

The manuscript of Ado and collaborators addresses the origin of the orbital Edelstein effect. In the manuscript, an induced orbital momentum density is derived for a scalar potential gradient. Two derivations are in fact made, an elementary one and a microscopic derivation. These give both a scaling of the effect as tau^3.

1- In ref. 8 it was shown that in order to have the OEE in a crystal the local point group of the atom should not have inversion symmetry. It would be appropriate to specify in how far the here proposed gradient of the potential is different from the mechanism of ref. 8. When there is no inversion symmetry in z direction, there is a difference of the potential in +z and -z direction. The here considered gradient seems to be a special and less general form of this condition.

2- In section 2.2 there is a subtle change performed from operators in the first equations macroscopic quantities such as the drift velocity. It might be correct, but the expectational value of zv_x is then the product of two quantities, which is dangerous. The authors should specify why their rewriting of <zv_x> is possible.

3- The final equation (12) or (13) is written to depend on tau^3. This might be so when one rewrites the mean free path l, but then a further E also appears. In the end, the equation has then M proportional to E^3, but this is a higher order correction to the OEE. Eq. (25) seems to be different, however.

4- The form of how the electric field is included might also warrant further explanation. It is done through the mentioned substitution of the drift velocity, whereas the applied electric field doesn't appear in the Hamiltonian. There is instead the magnetic field that is derived from the vector potential in the Hamiltonian. An explanation is appropriate here, since E would be the time derivative of the vector potential, but this is zero for the chosen form, i.e., no E field.

Requested changes

The authors should consider points 1 to 4 in the above section and address these.

Recommendation

Ask for major revision

  • validity: good
  • significance: good
  • originality: high
  • clarity: ok
  • formatting: good
  • grammar: excellent

Author:  Ivan Ado  on 2024-12-09  [id 5033]

(in reply to Report 1 on 2024-11-20)

We thank the Referee for reviewing our manuscript. Below we reply to the points raised in the review.

*0* We agree that our result for the orbital Edelstein effect (OEE) “does not explicitly depend on the materials’ specific electronic structure”. However, we do not think this is a weakness. Indeed, a finite scalar potential gradient can be hosted by different materials with different electronic properties (bulk Rashba materials is a particular example). In our work, we demonstrate that such a gradient leads to a finite OEE, thus qualifying the corresponding mechanism to be a universal property rather than a material specific one.

*1* Our model system is fully symmetric in the $xy$ plane and asymmetric with respect to the $z$ direction. Its point group is $C_{\infty v}$. Ref. 8 studies two noncentrosymmetric materials described by the $D_{4h}$ point group. $C_{\infty v}$ and $D_{4h}$ are two particular examples of point groups lacking the inversion symmetry operation.

We note that Ref. 8 does not specify which mechanism determines OEE in the studied materials and that the reported analysis is numerical. We also note that Ref. 8 considers magnetic materials while the obtained results depend on the orientation of the atomic magnetic moments. Our model system is nonmagnetic.

*2* We agree with the Referee that our “elementary derivation” practically replaces the average product by the product of averages. Such an operation is indeed not rigorous, but it does provide a clear intuitive interpretation of the studied effect. Since it also gives the correct result, as we show by performing a controllable microscopic derivation, we consider it important to report. Technically, we anticipate that the “elementary derivation” is allowed because of the absence of spin-orbit coupling in the system. In the new version of the text, we will discuss this matter in more depth.

*3* The mean free path is equal to the product of the Fermi velocity and the scattering time (or the transport time). None of these two factors depend on the electric field E, at least in the linear response. Thus, no additional powers of E appear. We also note that Eq. (12) and Eq. (25) are identical (we mention this in the text right after the latter).

*4* In the linear response theory, the relevant perturbation is “extracted” from the Hamiltonian at the start of the calculation. The subsequent analysis is performed with respect to the unperturbed Hamiltonian. In our case, this means that the electric field does not enter the Hamiltonian.

---

## Round 2 · Referee Report · Anonymous (Referee 2) · 2025-3-31

Report

In the present work the authors propose a way to induce an Orbital Edelstein Effect (OEE) in a three-dimensional electron gas with a potential $U(z)$ that breaks mirror symmetry $z\to -z$. The potential is proposed to be the source of a finite in-plane orbital magnetization $M_y\propto zp_x$. The wave function is a scalar and has no spin nor internal orbital structure. Upon expansion of the potential in powers of $z$, in a way that an electric field appears, $U(z)=U_0+z\frac{\partial U}{\partial z}+\ldots$, a finite in-plane magnetization results to be proportional to the electric field $\langle M_y\rangle \propto \partial U/\partial z$.

I find the derivation not convincing, tricky and full of loopholes. First of all, it is not clear whether the potential $U(z)$ is a confining potential, that confines the motion in the $xy$ plane or not.

In case it is not a confining potential and it only comprises an electric field $\partial U/\partial z$ (that is the only characteristics of the potential that appears in the entire derivation), then one should consider an electric field pointing in a general direction in the $xz$ plane, with $E_x$ and $E_z\propto \partial U/\partial z$ components. In this case there is only electron drift along the direction of the applied electric field.

In case $U(z)$ is a confining potential, then the origin of the system is arbitrary along the $z$ direction. For example, by introducing a fictitious magnetic field $B_y$ described by a vector potential $A_x=-B_y(z-z0)$, the resulting "orbital magnetic moment operator" would acquire an additional term proportional to $z_0$, and the latter could be chosen to be $z_0=\int dz z n(z)$, so to nullify the expectation value of $\hat{M}_y$.

Also, the introduction of this "orbital magnetic moment operator" is not convincing. What is well defined is the orbital magnetization as $-\partial F/\partial B_y$ (with $F$ the equilibrium free energy, I leave aside a discussion out of equilibrium, but analogous results should follow) in presence of an external magnetic field. The latter is associated to orbital currents and is typically calculated in linear response as a current-current response function, with paramagnetic and diamagnetic terms, and it goes to zero for zero field.

The operator $xp_z-zp_x$ is the angular momentum operator $L_y$ and, upon a rotation in the $xz$ plane such that the momentum is along the drift direction given by the combination of $E_x$ and $\partial U/\partial z$, an ill-defined expressions would appear, as already pointed out by the authors.

For these reasons I find the work not suitable for publication.

Recommendation

Reject

---

## Editorial Decision

awaiting_resubmission